# Correlation between CRISPR Loci Diversity in Three Enterobacterial Taxa

**DOI:** 10.3390/ijms232112766

**Published:** 2022-10-23

**Authors:** Dumitrana Iordache, Gabriela-Maria Baci, Oana Căpriță, Anca Farkas, Andreea Lup, Anca Butiuc-Keul

**Affiliations:** 1Doctoral School of Integrative Biology, Babeș-Bolyai University, 44 Republicii street, 400015 Cluj-Napoca, Romania; 2Department of Molecular Biology and Biotechnology, Faculty of Biology and Geology, Babeș-Bolyai University, 1 M. Kogalniceanu Street, 400084 Cluj-Napoca, Romania; 3Centre for Systems Biology, Biodiversity and Bioresources, Babeș-Bolyai University, 5–7 Clinicilor Street, 400006 Cluj-Napoca, Romania; 4Faculty of Animal Science and Biotechnology, University of Agricultural Sciences and Veterinary Medicine, 400372 Cluj-Napoca, Romania

**Keywords:** CRISPR-Cas, loci polymorphism, PCR primers, *Salmonella enterica*, *Escherichia coli*, *Klebsiella pneumoniae*

## Abstract

CRISPR-Cas is an adaptive immunity system of prokaryotes, composed of CRISPR arrays and the associated proteins. The successive addition of spacer sequences in the CRISPR array has made the system a valuable molecular marker, with multiple applications. Due to the high degree of polymorphism of the CRISPR loci, their comparison in bacteria from various sources may provide insights into the evolution and spread of the CRISPR-Cas systems. The aim of this study was to establish a correlation between the enterobacterial CRISPR loci, the sequence of direct repeats (DR), and the number of spacer units, along with the geographical origin and collection source. For this purpose, 3474 genomes containing CRISPR loci from the CRISPRCasdb of *Salmonella enterica, Escherichia coli,* and *Klebsiella pneumoniae* were analyzed, and the information regarding the isolates was recorded from the NCBI database. The most prevalent was the I-E CRISPR-Cas system in all three studied taxa. *E. coli* also presents the I-F type, but in a much lesser percentage. The systems found in *K. pneumoniae* can be classified into I-E and I-E*. The I-E and I-F systems have two CRISPR loci, while I-E* has only one locus upstream of the Cas cluster. PCR primers have been developed in this study for each CRISPR locus. Distinct clustering was not evident, but statistically significant relationships occurred between the different CRISPR loci and the number of spacer units. For each of the queried taxa, the number of spacers was significantly different (*p* < 0.01) by origin (Africa, Asia, Australia and Oceania, Europe, North America, and South America) but was not linked to the isolation source type (human, animal, plant, food, or laboratory strains).

## 1. Introduction

The CRISPR array was highlighted for the first time in 1987 [1], but the role of the CRISPR-Cas system was only grasped years later, in 2004 [2]. In the past, the system was reported to be present in 85–90% of archaea and 45–50% of bacteria [3,4], but, since those percentages reflect only the situation in the available databases and new genomes are constantly sequenced, these values may vary over time. To date, as of April 2022, in the CRISPRCasdb [5], part of the CRISPRCas++ website [6], only 60.94% of archaea and 34.24% of bacteria are reported to contain both a cluster of Cas proteins and at least one CRISPR array. CRISPR-Cas is a bacterial defense system against foreign genetic material [2,7]. Briefly, the CRISPR-Cas system consists of two main genetic elements: the CRISPR arrays and CRISPR-associated (*cas*) genes. The structure of the CRISPR arrays implies two types of sequences, specifically, highly conserved sequences, known as direct repeats (DR), and sequences called spacers that are derived from exogenous DNA or RNA [8,9]. CRISPR-Cas adaptive immunity consists of three stages: The first stage, called adaptation, represents the incorporation of spacer sequences into the CRISPR loci each time the cell encounters new and foreign genetic material. The second step, expression, involves the transcription of the CRISPR array and the processing of pre-crRNA into mature crRNA. The third step comprises the interference of the exogenous genetic material and mature crRNA, followed by the degradation of the complex. Several Cas proteins mediate the process, which has been previously reviewed by numerous authors [10,11,12,13,14,15]. The diversity of the CRISPR-Cas system is relatively high, depending on the *cas* genes; two classes, six types, and 33 subtypes have recently been described [16]. 

The successive addition of spacer sequences in the CRISPR array is a mechanism for inducing genetic variability in the prokaryote genome, which has made the CRISPR-Cas system a valuable molecular marker for the structure of microbial populations [17], bacterial genotyping [18], phylogeny [19] and epidemiology [20,21,22]. Thus, a quick, easy, and accurate method for the identification of the system is required for such multiple applications. Due to the most significant characteristic of the system, their repeating units being separated by unique insertions of similar lengths, the CRISPR array is easily identifiable in long sequences of DNA [23]. However, because of the repetitive nature of CRISPRs, their analysis in metagenomic data is difficult, as CRISPR loci do not typically assemble [24]. Currently, next-generation sequencing (NGS) is being used to identify the CRISPR-Cas systems, followed by the employment of software developed to search for these loci [25,26]. Various algorithms have been developed to identify CRISPR arrays. In the past, the most commonly used ones were CRISPRFinder, PILER-CR, and CRT. One of the most popular algorithms is CRISPRCasFinder, from CRISPRCas++ [6], which represents an enhanced version of CRISPRFinder [27] that also integrates tools for searching Cas proteins [28,29]: Prodigal recognizes and translates the coding sequences, while the MacSyFinder identifies the families of Cas proteins using the Hidden Markov Model to determine the similarities between the different Cas proteins encoded by these genes, then groups them into families [28,29]. In addition, the authors applied their algorithm to create a database. Its source is GenBank, and it analyses all genomes labeled as either complete genomes or chromosomes [5]. 

NGS provides higher discovery power to detect novel genes and higher sensitivity to quantify rare variants and transcripts. A single NGS experiment can identify variants across thousands of target regions with a single-base resolution. NGS allows a shallow scan across multiple samples or a sequence at greater depth, with fewer samples, to find rare variants in each region [30,31]. Although genotyping by NGS has reduced the time and cost of sequencing data, compared with other sequencing methods, it is still expensive for most labs, which limits its widespread application or the analysis of a large number of samples. Moreover, several applications tend to be confirmed using Sanger sequencing methods because the next-generation methods have issues with copy number variants [32]. Other technological and biological aspects of NGS that are essential to consider are the file formats, alignment tools, genomic browsers, and visualization methods for comparative genomics. With the expansion of sequencing data and different software companies developing methods for using these datasets, archiving genomic data needs to be standardized [33].

The polymerase chain reaction (PCR) could offer an alternative method, due to its simplicity, low cost, familiar workflow, and capital equipment already being placed in most labs [31]. In the last years, many reference genomes have been available in databases, thereby making possible the development of highly efficient PCR primers to target a wide range of CRISPR arrays from different taxa. 

*Enterobacteria*, having a high diversity of species that are involved in many diseases in humans and animals, were previously analyzed regarding the structure of the CRISPR-Cas system. The most encountered CRISPR type is I-E, followed by the I-F type, and very few contain three different types [34]. Shen et al. [35] have identified two variants of the I-E Cas-cluster, I-E and I-E*, based on the differences in Cas1 and Cas3 regarding their sequences and their positions in the chromosome. In this article, we aimed to perform an advanced characterization of the CRISPR loci from three different *Enterobacteria* taxa—*Salmonella enterica, Escherichia coli* and *Klebsiella pneumoniae,* to establish the correlation between the enterobacterial CRISPR loci, the DR sequence, and the number of spacer units with a geographical origin and collection source, and to design PCR primers for each CRISPR locus from these taxa. By analyzing a much larger number of isolates than in previous studies, from all the phylogenetic groups and in all types of environments, we tried to enhance the primer’s target recognition by selecting those regions with minimal substitutions between the flanking sequences of the arrays. 

## 2. Results

### 2.1. Classification of CRISPR Loci in S. enterica, K. pneumoniae and E. coli Genomes

In this study, we have inspected 3474 genomes of *Enterobacteria*, available in CRISPRCasdb, covering a total of 6562 CRISPR loci from three taxa: *Escherichia coli, Salmonella enterica,* and *Klebsiella pneumoniae*. All queried *S. enterica* and *K. pneumoniae* genomes possess only one type of CRISPR-Cas system in their chromosome—I-E. By analyzing the sequences of Cas1 and Cas3, it is clear that the system containing the Kp1 locus belongs to the I-E type, being more similar to those of *E. coli* and *S. enterica*, while the system containing the Kp2 and Kp3 CRISPR loci belongs to the I-E* type. In the queried *E. coli* genomes, 94.57% of the detected Cas clusters were classified within I-E, with 5.42% classified into the I-F type (Figure 1). None of the genomes has more than one Cas-cluster of the same type, except for *Escherichia coli (E. coli)* SCAID URN1-2019, which exhibits two sets of CRISPR I-F loci. In this particular case, most of the sequences that flank the loci correspond to those identified for the Ec3 and Ec4 loci, except for one—the upstream sequence of one of the Ec3 loci is different. 

All the CRISPR-Cas systems are found on the chromosome, except for one CRISPR locus in *K. pneumoniae* that is located on a plasmid, Kp4-Pl. This was determined to be type IV, even if there was no Cas cluster detected.

### 2.2. Frequency of CRISPR Loci among the Surveyed Enterobacterial Genomes

The proportion of strains carrying each locus type is represented in Figure 2. According to the CRISPRCasdb, most of the queried *S. enterica* and *E. coli* genomes possess CRISPR loci (∼98% and ∼90% of the total number of genomes, respectively) in contrast to the substantially lesser *K. pneumoniae* genomes that are positive for CRISPR loci (∼30%) (Figure 3). The number of genomes identified as having the CRISPR-Cas systems is approximately 4–10% lower than genomes containing CRISPR arrays. The presence of the type I-F CRISPR-Cas system identified in *E. coli* is very low (4.36% of genomes) compared to type I-E (∼76%). We found that of the total number of genomes containing the I-F system, 26.88% also contain the I-E system, while in the case of the genomes containing Ec3b (the orphan locus assigned to I-F, due to the DR sequence), 89.76% contain the I-E system, although this locus is most probably inactive since it lacks the Cas-cluster and it has a very low number of spacers.

### 2.3. Characterization of Repeat/Spacer Units

The spacer sequences in both the I-E and I-F CRISPR loci consist of 32 nucleotides, with very few exceptions that have an extra one. The most frequently encountered direct repeat sequence is G**T**GTTCCCCGCGCCAGCGGGGATAAACCG (DR1a)—present in all three taxa—both loci from *S. enterica*, Ec1, Ec2a, and Kp1, while most of Ec2b have adenine instead of thymine in the second position (DR1b). The repeat sequence for Kp2 is GTCTTCCCCAC**A**C**G**CGTGGGGGTGTTTCT (DR2a), whereas the Kp3 repeat differs from Kp2 by two transition substitutions in the 12th and 14th positions (DR2b). DR1 and DR2 also have other variants that are far less prevalent and can be found in Appendix A. The repeat sequence for the type I-F loci—Ec3a, Ec3b, and Ec4—is GTTCACTGCCGTACAGGCAGCTTAGAAA (DR3). The number of repeat/spacer units varies between the CRISPR arrays, even for the same locus (Figure 4).

In the isolates possessing the CRISPR loci, the average number of spacers per genome was 34.09 for *S. enterica*, 22.67 for *E. coli,* and 27.24 for *K. pneumoniae*, respectively (Table 1). We used the annotation locus/DR for the statistical analyses of those loci presenting a specific direct-repeat sequence. A total of 1280 genomes of *S. enterica* were found to carry CRISPR loci, with an average of 15.46 spacers for Se1/1a, 8.00 for Se2a/1a and 23.38 for Se2b/1a. In the 1901 genomes of *E. coli* possessing CRISPR loci, the ones that were most frequently found were Ec2a/1a, Ec1/1a, and Ec2b/1b, with an average number of spacers of 5.94, 5.24, and 2.21, respectively. CRISPR loci were identified in 293 *K. pneumoniae* genomes, with the highest number of spacers observed for the loci Kp1/1a, Kp2/2a and Kp3/3b, with average values of 16.17, 6.78, and 3.06, respectively (Appendix A).

Statistically significant relationships have been identified between the presence of different CRISPR loci and the number of spacer units, according to geographical origin and isolation source types. For each of the queried taxa, the number of CRISPR spacers was significantly different (*p* < 0.01, Kruskal–Wallis test) when assessed according to origin (Africa, Asia, Australia and Oceania, Europe, North America, and South America). However, the difference between the number of CRISPR spacers was not statistically significant (*p* > 0.5) when assessed according to the collection site (human, animal, plant, food, or laboratory strains). 

For *S. enterica*, the overall statistically significant difference by location was due to the genomes originating in Europe and South America. *S. enterica* from Europe was found to possess the lowest number of CRISPR spacers (on average, 18.19 per genome), thus being significantly different from those isolates originating in Asia, North America, South America, Australia and Oceania, and Africa (*p* < 0.001, Dunn’s test). On the other hand, *S. enterica* genomes from South America were found to possess the highest number of CRISPR spacers (on average, 47.86 per genome). They differ significantly from the genomes from other continents (*p* < 0.05), except for Australia and Oceania. For *E. coli*, the overall statistically significant difference by location was due to the genome originating in North America, with the highest number of spacers (an average of 31.55 per genome), differing significantly from the European and Asian isolates (*p* < 0.001, Dunn’s test) but also from those collected in Africa and South America (*p* < 0.05). For *K. pneumoniae*, the overall statistically significant difference was due to the genomes originating in Asia (on average, 26.96 spacers per genome), which differ significantly from those genomes isolated from Europe and North America (*p* < 0.05) but not from those isolated from Australia and Oceania. 

In *S. enterica*, negative correlations were found between the CRISPR loci Se1 and Se2c (r = −0.720). A weak negative correlation overall between the number of spacer units was found for the CRISPR variants Se1/1a and Se2c/1a (r = −0.295), with the highest values in *S. enterica* genomes originating from Asia (r = −0.414) and being isolated from human sources (r = −0.453). The overall negative correlation between the number of spacer units for CRISPR DR Se2a/1a and Se2b/1a (r = −0.374) was moderate in the *S. enterica* genomes collected from South America (r = −0.689) and in the environmental isolates (r = −0.553). Negative correlations were observed between the number of spacers in loci Se2b/1a and Se2c/1a, only in the Australian (r = 0.503) and European genomes (r = −0.325) according to location and in human isolates (r = −0.307) according to host. When considering the number of spacers in each CRISPR locus, an overall positive weak correlation was found between the loci Se1/1a and Se2b/1a (r = 0.383), which was moderate in *S. enterica* from Africa, found in environmental and food isolates. There was a lack of correlation between these systems in Australian isolates and in *S. enterica* isolated from vegetation. In addition, positive correlations were found between Se1/1a and Se2a/1a in Australia, South America, and Asia, in isolates from plant specimens, and between Se1/1d and Se2a/1a in North American isolates (r = 0.491). 

In *E. coli*, a very strong positive correlation was found between the presence of the I-F CRISPR loci, Ec3a and Ec4 (r = 0.994), and also between their numbers of spacer units: Ec4/4a and Ec3a/4a (r = 0.906), with the highest values in *E. coli* genomes originating in Asia (r = 0.972) and in those isolated from animal sources (r = 0.925). Negative correlations were found between loci Ec1 and Ec3a/Ec4. Assessing their DR sequence in the Ec1 locus, the overall negative correlation between spacer units of Ec1/1a and Ec1/1b (r = −0.409) has been moderate in *E. coli* isolates originating in Africa (r = −0.655) and isolated from food sources (r = −0.540). The number of spacers in the locus Ec1/1k was weakly correlated with the number of spacers in locus Ec2a/1k (r = 0.414). This positive correlation was moderate in those genomes isolated from Asia (r = 0.504) and in those with a human host (r = 0.585). When assessing the number of spacers, the weak overall negative correlations found between the loci Ec1 and Ec3a/Ec4 could not be assigned to any particular combination of CRISPR variants, neither by geographic origin nor by isolation source types. An overall weak correlation was found between the number of spacers in the Ec2 locus variants Ec2a/1a and Ec2b/1b (r = −0.374), with the highest values in those *E. coli* isolates of geographic origin in Asia (−0.428) and isolated from food sources (−0.393). Negative weak correlations were found between Ec2a/1a and Ec2a/1e in Africa (−0.318), Ec1/1b and Ec2b/1b in Africa (r = −0.289) and Australia and Oceania (r = −0.256), and between Ec1/1a and Ec2a/1e in Australia and Oceania (r = −0.256). 

In *K. pneumoniae*, a strong positive correlation was found in the presence of CRISPR loci belonging to the same system, Kp2 and Kp3 (r = 0.937), while negative correlations were observed between Kp1 and Kp2/Kp3 (r = −0.786/−0.825) since those systems do not coexist in the same genomes. Overall, strong positive correlations between the number of spacer units were found for Kp2/2a and Kp3/2b (r = 0.536), with the highest values in the European isolates (r = 0.638) and those isolated from *Homo sapiens* (r = 0.556). Positive correlations occurred between Kp2/2a and Kp3/2c (r = 0.340), observed mainly in North American (r = 0.616) and also in European and Australian isolates, being very strong in environmental *K. pneumoniae* (r = 0.992) and also between Kp2/2a and Kp3/2d (r = 0.315), being more evident in human isolates from Asia and Australia (see Appendix A). 

In the principal component analysis (PCA), no distinct clustering was observed by geographical origin or isolation source types for any of the enterobacterial taxa. For *S. enterica*, most of the genetic variation was explained by the first two principal components among locations (86.45% of the total variation) (Figure 5a) and also among sources (88.42%) (Figure 5b). For *E. coli*, the first two principal components contributed to 66.68% of the total variation concerning geographical origin (Figure 5c) and to 63.11% for isolation sources (Figure 5d). For *K. pneumoniae*, the first two principal components accounted for 93.48% of the various locations (Figure 5e) and 94.06% among source types (Figure 5f). Furthermore, for all the three species investigated during this study, the multivariate nonparametric permutational analysis of variance (PERMANOVA) revealed significant differences (*p* < 0.001) between populations according to geographical origin, as explained by differences in the dispersion of CRISPR elements (Welch’s F-test, *p* = 0). When assessing the variance between populations according to the isolation source, no differences were observed in the variances of CRISPR elements for *E. coli* and *S. enterica*. Nevertheless, although the PCA indicates a similarity in their CRISPR system diversity, environmental and human-derived *K. pneumoniae* strains differ significantly (PERMANOVA, *p* < 0.001), due to the heterogeneity of variance. The PCA analysis simplifies the high-dimensional data, offering an integrative view of the CRISPR components involved in variation. Thus, the great diversity of CRISPR patterns was reduced in dimensionality, and data interpretation increased. In *S. enterica,* the major loadings were set by Se2b/1a, Se1/1a, and Se2a/1a for PC1, and Se2a/1a and Se1/1a for PC2, respectively. For *E. coli,* Ec1/1a and Ec2a/1a had major loadings for PC1, while Ec2a/1a and Ec2b/1b were the major contributors for PC2, respectively. In *K. pneumoniae*, Kp1/1a and Kp2/2a were the major contributors in PC1, while Kp2/2a and Kp4/3a were the major contributors in PC2, respectively.

### 2.4. PCR Primers for CRISPR Loci

The pairs of primers for the CRISPR loci and the alignment temperature for each pair are shown in Table 2. For those loci with more than one variant, the primers can be used together—the reverse and the two or three forward sequences, since none of the genomes have more than one variant. In silico testing showed very high specificity for all the primers. For each species, the primers have been tested in silico on 100 isolates that are positive for CRISPR, selected randomly from the database. The CRISPR loci have been correctly identified in a proportion of 100 percent. 

## 3. Discussion

In *E. coli* and *S. enterica,* the I-E systems have a CRISPR locus on each side of the Cas-cluster, which is also the case for I-E* in *K. pneumoniae*; however, the I-E system in *K. pneumoniae* has only one CRISPR locus, upstream of the Cas-cluster. Several genomes seemingly lack some units (CRISPR loci or Cas-cluster) from the typical conformation. As a result, the number of CRISPR loci associated with the same cluster, even though they are similar, can differ. The CRISPR loci and the Cas-cluster in the same system have the same orientation, but the two systems in *E. coli* have opposite orientations. Knowing the correct orientation for each locus, all the DR sequences reported in this study are forward, while in previous studies, the orientation of the DR was not considered [36,37,38]. 

The CRISPRCasFinder algorithm did not identify any Cas proteins for the level-4 CRISPR locus on the plasmid, Kp4-Pl, although there was a locus that fits all the criteria for Kp4-Pl that was classified as level 3, with a Cas-cluster identified as type IV. In addition, the sequences of plasmids where level-4 CRISPR arrays were found were run through the CRISPRone program [39] using the default criteria. The software considered the arrays to be most probably of type IV, even though they do not have the typical conformation, only some of the proteins being found. Classification of the locus Kp4-Pl as being type IV was also suggested by Kamruzzaman and Iredell [40]. They concluded that the conformation of the Cas-cluster determined by the CRISPRone software is highly similar to one found in the type IV Cas-cluster from *Aromatoleum aromaticumi* EbN1, with only one subunit from the conformation being missing in *K. pneumoniae*. Regarding the type IV locus, what was somewhat surprising was the lack of a positive correlation between Kp4-Pl and any other CRISPR-Cas system, since it had been speculated before that this type might be dependent on another system for its competence [40]. Even more, a significant negative correlation was found between Kp1 and Kp4-Pl (−0.265), and though not significant, there was still a negative correlation between Kp4-Pl and KP2/Kp3 (−0.215/−0.232).

In *E. coli,* the two types of systems (I-E and I-F) are not usually found in the same genome [36,41,42], but there is an orphan locus—Ec3b—that has a DR sequence that matches that of the I-F system, and it is a variant of the first locus of I-F—Ec3a. In the genomes containing this variant, the I-E system is often encountered. Considering that the upstream region of the Ec3b locus is identical to the upstream region flanking the CRISPR-Cas I-F system, that it has a minimal number of spacers and that it never coexists with the I-F system, this locus may be a remnant of the I-F system that has been lost.

According to the literature, in *E. coli*, there appears to be a correlation of the distribution of the spacer number with pathogenic traits [36,37,41]. The combined distribution of spacer numbers between the two groups is similar to the distribution presented in this study, which integrates all *E. coli* genomes, regardless of their pathogenicity and environments. However, for *S. enterica* and *K. pneumoniae*, a small number of genomes were analyzed in previous studies [20,38], and, for this reason, the maximum number of spacers reported is much lower than the number currently reflected in the database, as shown in our study.

The CRISPR-Cas system has been previously used in *S. enterica* to determine the CRISPR loci polymorphism and to differentiate between the different serotypes in case of outbreaks. Two CRISPR loci have been identified [20,21] and the different constructions of the system have been attributed to a few serovars [20]. Although the spacer composition has been reported to be correlated with the *S. enterica* serotypes, we are unable to determine such a relationship in our work; in the case of most of the genomes included in this study, the serotype information is not provided in the database that we used. Previous studies also developed primers to amplify the CRISPR loci from *S. enterica* [18,43], *E. coli* [36,37], and *K. pneumoniae* [44,45], by targeting the adjacent genes of the system and the *cas* genes. Since then, a large number of genomes have been added to the public genome repositories for these species. It is important to note that the primers designed in this study were based on a larger number of genome datasets than those from past studies, allowing us to design the primers from directly adjacent regions of the CRISPR loci, enabling the identification of polymorphism occurring in each system. It would be of future interest to infer the serotypes and/or sequence types of the bacterial strains, employing the relevant genome-based typing methods prior to extracting the CRISPR systems using our highly efficient primers, to allow further exploration of their potential role as a molecular epidemiological marker.

## 4. Materials and Methods

### 4.1. Gathering Data

For this study, we have inspected the genomes of *Enterobacteria*, available in CRISPRCasdb, from three taxa: *Escherichia coli, Salmonella enterica,* and *Klebsiella pneumoniae*. The database was implemented using the algorithm CRISPRCasFinder, which possesses a rating system that distinguishes between false CRISPR elements and real CRISPR arrays. We utilized an Ebcons index (entropy-based conservation index), based on an algorithm that allows quantifying the level of conservation of DR and spacers, and it includes four levels of evidence [29]. However, here only level 4 CRISPR loci have been taken into consideration, which involves repeat EBcons ≥ 70 and the percentage of identity regarding spacers—at most, 8%.

### 4.2. CRISPR Loci Analysis

The main CRISPR loci for each taxon have been determined by their flanking sequence and their position relative to the Cas-cluster. Several aspects have been considered for each CRISPR locus: the family to which they belong, the presence/absence of the Cas cluster, and the CRISPR orientation. The program automatically annotates the family of the Cas-cluster. Thus, the family of the CRISPR locus can be identified by affiliation with a cluster of *cas* genes. An important aspect to consider is the CRISPR orientation, which helps us to determine the position relative to the Cas-cluster and other CRISPR loci correctly, but most importantly, to compare the CRISPR loci from different isolates. Another feature that was noted was the repeat sequence, which can help to identify the CRISPR family when the Cas-cluster is missing and establish the transcription direction where the program could not find it. The number of spacers for each CRISPR locus was indicated as well.

### 4.3. Primers Design

For the design of PCR primers, the sequences flanking the loci were analyzed—with 100 nucleotides on the upstream and downstream. The upstream flanking sequence of a CRISPR includes the leader sequence, this being rich in AT, while the downstream one contains a relatively equal proportion of AT and CG. Similar sequences were aligned, and regions that did not differ between each sequence were selected. Primers were designed using FastPCR [46], with a sensitivity level of 3 being selected. The alignment temperature was calculated using the Tm calculator from Thermo Fisher Scientific [47], for a Taq-based DNA polymerase. 

### 4.4. Statistical Analysis

For statistical analysis, the CRISPR-bearing genomes were allocated to groups based on either their geographical location or isolation source. Six regional groups were created to reflect the origin of bacteria: Africa, Asia, Australia and Oceania, Europe, North America, and South America. In addition, seven groups were set up by the source of specimen: human, animal (poultry, swine, sheep, equine, cattle, wild birds, fish, reptile, rodent), food (meat, milk, seafood, eggs, food, animal feed), plant, environment (water, wastewater, soil), laboratory and missing host. Genomes with missing data on country of origin and isolation source or host and groups with a low sample size were excluded from statistical comparisons. For each enterobacterial species, correlations between groups were assessed to compare the presence of different CRISPR loci and the frequencies of CRISPR spacers in specific arrays. To assess significant differences by location and collection site, we compared the abundances of the CRISPR spacers between groups using the non-parametric Kruskal–Wallis test and Dunn’s post hoc test, undertaken with a minimum α level of 0.05, with a Bonferroni correction. Correlations and analysis of variance were conducted using the Real Statistics Resource Pack software (Release 7.6). PCA was executed for the clustering and differentiation of data sets by PAST software, version 4.11 [48]. The PCA multivariate statistical approach was used to reduce the multidimensional CRISPR datasets and to explore the inter-correlations to a smaller set of principal components (PCs). Direct and indirect associations of specific CRISPR systems with PCs best described the variance that was discussed. To assess whether the CRISPR-bearing bacterial populations differ between continents or sources, PERMANOVA was run with 99,999 permutations using the PAST software. The assumption of homogenous multivariate dispersion was assessed using the Levene’s test. Under the assumption of heteroscedasticity (Levene’s test), a distribution-based Welch’s F-test was employed to examine if variability is due to dispersion differences among groups.

## 5. Conclusions

The polymorphism of CRISPR loci, mainly the DR sequence and the number of spacers, were identified in the enterobacterial genomes. Significant differences between spacer sequences and the origin (Africa, Asia, Australia and Oceania, Europe, North America, and South America) of the investigated taxon were found. However, there is no connection with the isolation source type (human, animal, plant, food, or laboratory strains). 

The most prevalent CRISPR type is I_E, and the repeat sequence is similar or identical between the CRISPR loci. Nevertheless, type I_F was also identified in *E. coli* genomes containing type I_E, which was previously reported as highly unlikely.

Specific primers targeting the CRISPR loci in *Salmonella enterica, Escherichia coli* and *Klebsiella pneumoniae* were developed and the validation in silico demonstrated their high efficiency.

## Figures and Tables

**Figure 1 ijms-23-12766-f001:**
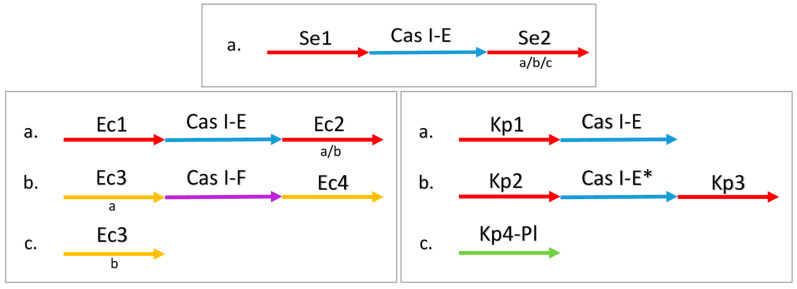
The typical conformation of CRISPR-Cas systems in each taxon. Each configuration (a, b, c) is independent, they do not need to co-exist in the same genome. All the arrays from a CRISPR locus have the same position relative to the Cas cluster (upstream or downstream) along with the same 5′ end-flanking sequence and DRs sequence; those with different 3′ end-flanking sequences are considered different variants of the same locus. The variants are listed below the arrows, in lower-case letters. Ec3a and Ec3b have the same upstream but different downstream sequences. The isolates of *S. enterica* and *K. pneumoniae* contain only one Cas-cluster (I-E), while *E. coli* isolates may have one or two Cas-clusters (I-E and I-F).

**Figure 2 ijms-23-12766-f002:**
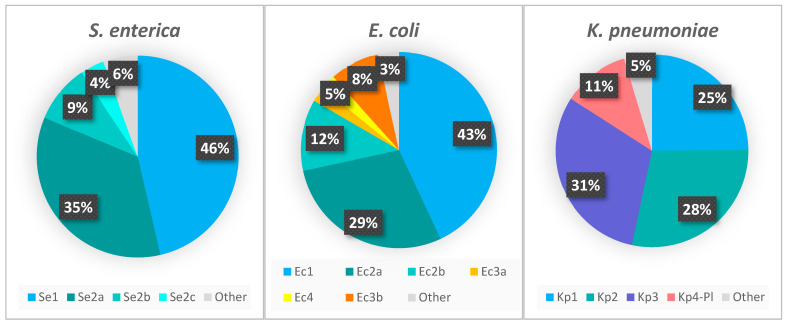
The proportion of CRISPR loci, regardless of the presence of the Cas-cluster. “Other” represents the CRISPR arrays that are not part of the main loci.

**Figure 3 ijms-23-12766-f003:**
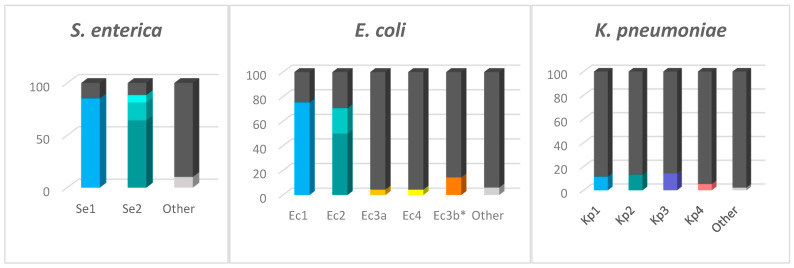
Genomes containing the CRISPR loci. The bars represent the total number of genomes investigated. Colors were used to represent the genomes containing the CRISPR loci, while the dark gray is used to represent the genomes lacking each specific locus. The different shades from Se2 and Ec2 represent variants of the same locus. The Ec3a and Ec3b, even though they have identical sequences at the 3′ end, are represented here in separate bars because Ec3b is always an orphan locus and never replaces Ec3a in the Ec3a-Cas-Ec4 system. “Other” represents the CRISPR arrays that are not part of the main loci.

**Figure 4 ijms-23-12766-f004:**
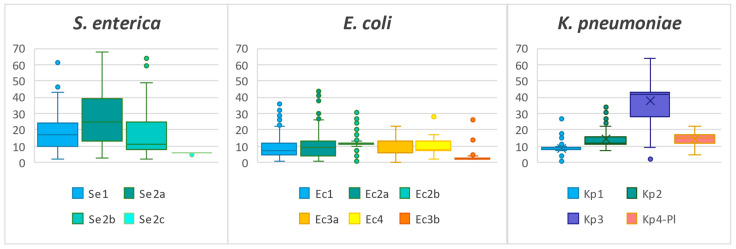
Distribution of the number of repeat/spacer units for each locus. The Se2a locus contains an outlier of 124 units that is not shown.

**Figure 5 ijms-23-12766-f005:**
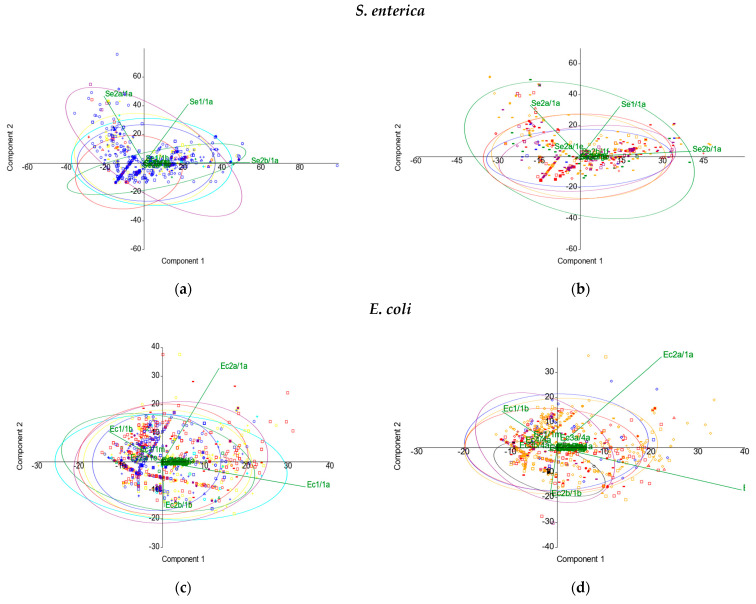
PCA analysis by (**a**,**c**,**e**) geographical origin and (**b**,**d**,**f**) isolation source types.

**Table 1 ijms-23-12766-t001:** Distribution of CRISPR spacers in the queried genomes, according to geographic origin and source of isolation.

	Continent of Origin						Source of Isolation					
	Africa	Asia	Australia/Oceania	Europe	North America	South America	Plant	Environ mental	Human	Animal	Food	Laboratory
*S. enterica*												
N	15	202	14	153	821	21	38	30	355	291	198	5
(%)	1.22	16.48	1.14	12.48	66.97	1.71	4.19	3.31	39.14	32.08	21.83	0.55
Average	32.33	35.85	37.57	18.20	32.75	47.86	39.37	31.27	32.74	32.30	32.97	58.4
Median	36	34	35.5	7	27	49	29	25	34	26	24.5	55
SD	21.72	20.33	18.24	17.85	19.98	18.90	24.26	14.99	19.34	18.93	18.30	21.53
Minimum	6	2	6	3	2	9	4	9	3	3	2	31
Maximum	80	87	66	86	140	94	92	78	87	94	87	86
*E. coli*												
N	17	376	39	488	465	24	2	82	394	534	33	11
(%)	1.21	26.69	2.77	34.63	60.33	1.70	0.19	7.77	37.31	50.57	3.13	1.04
Average	19.12	21.08	23.62	19.89	31.55	20.79	18	20.11	18.95	21.53	16.91	20.64
Median	19	18	24	18	30	17.5	18	17.5	17	20	15	18
SD	8.58	11.32	11.53	10.78	21.00	13.63	0	11.01	11.39	11.30	7.70	10.81
Minimum	2	2	5	2	2	2	18	2	2	2	4	14
Maximum	33	65	52	66	140	48	18	53	65	66	33	53
*K. pneumoniae*												
N	3	142	20	41	43	2	0	10	194	5	0	0
(%)	1.20	56.57	7.97	16.33	17.13	0.80	0	4.78	92.82	2.39	0	0
Average	27	26.96	30.15	31.39	33.42	14.5	-	25.8	28.62	25.4	-	-
Median	22	21	25	35	33	14.5	-	24	24	17	-	-
SD	24.88	13.26	15.97	12.39	13.41	7.78		17.97	12.58	15.52		
Minimum	5	2	7	8	6	9	-	2	5	13	-	-
Maximum	54	64	64	64	64	20	-	64	64	51	-	-

**Table 2 ijms-23-12766-t002:** PCR primers for CRISPR loci.

Locus	Primers	Ta (°C)	Elongation Time ^1^
Se1	f: 5′-AAATTGTTGCGATTATGTTGGTAG r: 5′-CTGGTACACAGATTATGATTATGC	55.0	3 min
Se2a	f: 5′-AAAGTTGGTGGGTTTTTTGTGC r: 5′-ATGCTGCCGTTGGTAAAAGAG	59.9	3 min
Se2b	f: 5′-ATAATGCTGCCGTTGGTAAAAGG r: 5′-ATGCTGCCGTTGGTAAAAGAG	59.9	3 min
Se2c	f: 5′-GGAAAAGTTGGTGGGTTTTTTG r: 5′-ATGCTGCCGTTGGTAAAAGAG	59.0	45 s
Ec1	f: 5′-CTCTTTAACATAATGGATGTGT r: 5′-CTTGAGAAAGAGATAACGGG	52.3	2.25 min
Ec2a	f: 5′-ATTGTTGCGATTATGTTGGTAG r: 5′-TTGATGGGTTTGAAAATGAGAG	55.3	2.25 min
Ec2b	f: 5′-ATGTTACATTAAGGTTGGTGGG r: 5′-TTGATGGGTTTGAAAATGAGAG	56.3	2.25 min
Ec3a	f: 5′-TTAACAACGGGCTAAACGTG r: 5′-AATGGTTTGAAGTTGAGAGTG	55.5	3 min
Ec4	f: 5′- AAAAAGGGTTTGAATCTGCG r: 5′-CTGATGGGCGAAGAGAAAG	54.1	3 min
Ec3b	f: 5′-TATTAACAACGAGCTAAACGTG r: 5′-CCCCTCACCGTCATATTTAA	54.1	3 min
Kp1	f: 5′-TTGTCCACTAACGTTATCGA r: 5′-GTAGCGATATTTATTCTCCGC	53.3	3 min
Kp2	f: 5′-GTTAAACTCTCGCTCTTTCAC r: 5′-ATTCCGCTTATTGCTAAGTCC	56.3	1.5 min
Kp3	f: 5′-CCGCTAAACACAATATGCTG r: 5′-CTTGTCGTCACTTGAAAGG	53.1	1.5 min
Kp4-Pl	f: 5′-AGTCCCATCTGCTTGTAGG r: 5′-TTTGATTTCACTGCCCGCT	56.9	1.5 min

^1^ Elongation time is calculated by considering the highest number of spacers for each locus.

## Data Availability

Not applicable.

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
