# Peer review of "Correlation between CRISPR Loci Diversity in Three Enterobacterial Taxa"

_ijms, 2022, doi:10.3390/ijms232112766_

Round 1
Reviewer 1 Report
The study investigated the correlation between the CRISPR loci, the direct repeats (DR) sequence, and the number of spacer units 102 with geographical origin and collection source and to design PCR primers for each locus from three enterobacterial taxa, E.coli K. pneumoniae and S. enterica. Althoguh the results could be interesting and important in the field, several revisions are required as listed below
in the results section of the absctract mor einformation about the three studies species could be included
line 75-81 add references, as well as ijn the line 89-93
improvement of the resolution of both figures 4 and 5 would be appreciated
is it possible to add percentacegs in the table 1?
any correlation with antimicrobial reistsance shoudl be discussed
line 119-123 should be place in the discussion, althoguh there is already a similar sentence, hence could be deleted
is it possible to include in the table 2 the size of targeted products with primers identified?
in the discussion, often there is presentation of results that were not shown in the dedicated section, so the discussion section could be rearranged in a proper way
conclusions should be included, underlining the impact of the study findings
Author Response
We would like to thank all the reviewers for their suggestions on our manuscript. The answers to the reviewer's questions are detailed further. All the corrections are marked with track changes in the manuscript.
Answers to reviewer comments
Reviewer 1
The study investigated the correlation between the CRISPR loci, the direct repeats (DR) sequence, and the number of spacer units 102 with geographical origin and collection source and to design PCR primers for each locus from three enterobacterial taxa, E.coli K. pneumoniae and S. enterica. Althoguh the results could be interesting and important in the field, several revisions are required as listed below
in the results section of the abstract more information about the three studies species could be included
We consider that the main information about the CRISPR-Cas systems from the three taxa was indeed missing from the abstract, and we tried to include it without exceeding too much the maximum number of words suggested for the abstract.
line 75-81 add references, as well as in the line 89-93
References were introduced.
improvement of the resolution of both figures 4 and 5 would be appreciated
Figure 5 was replaced with a vector format, which is resolution-independent, it can be scaled to any size without losing quality. Figure 4 is a chart imported directly from Microsoft Excel, so its resolution cannot be adjusted.
is it possible to add percentages in the table 1?
Yes, percentages were included as suggested.
any correlation with antimicrobial resistance should be discussed
The possible implication of the CRISPR-Cas system in antimicrobial resistance was not the aim of our study. Therefore, we had no results/discussion on this subject. Moreover, the possible connection between CRISPR-Cas system activity and antimicrobial resistance was not yet demonstrated, several studies support the idea that CRISPR-Cas system prevents resistance acquisition, but others do not. It depends on a large scale on the study, pathogen and other parameters.
line 119-123 should be place in the discussion, although there is already a similar sentence, hence could be deleted
We have deleted the lines from the results section, although we inserted in the description of Figure 1 the explanation for what we mean by a CRISPR locus “variant”.
is it possible to include in the table 2 the size of targeted products with primers identified?
The size of the products differs according to the number of spacers contained in the CRISPR array; thus it will be different for each product. The distribution of the number of repeat/spacer units for each locus is represented in figure 4.
in the discussion, often there is presentation of results that were not shown in the dedicated section, so the discussion section could be rearranged in a proper way
Amends have been made – several sections from discussions were moved to results.
conclusions should be included, underlining the impact of the study findings
We agree and the conclusions were added.
Reviewer 2 Report
In the study entitled “ Correlation between CRISPR loci diversity in three enterobacterial taxa”, the authors aimed to examine the diversity of the CRISPR loci present in Klebsiella pneumoniae, Salmonella enterica and Escherichia coli. While the authors stated that the number of CRISPR spacers differed significantly between strains of different geographical origins for each species included in this study, no analysis was further performed to look at the distribution and composition of these CRISPR arrays in different serotypes or phylogroups of these individual species. This would help to better understand the evolution of CRISPR-Cas systems and to explore if CRISPR array could act as a potential molecular marker for any particular ST or phylogroup, improving study significance and the usefulness of the designed primers. Therefore, it is my honest opinion that this study would still require significant amount of work before it can be considered for publication. Please see additional comments below.
1. In the first part of the results section, it is rather difficult to understand the structure and the organisation of the CRISPR-Cas systems present in these three bacterial species and Figure 1 is just too simple. What does it mean that the position relative to the Cas-cluster is the same for all variants? Were these CRISPR/Cas systems always found downstream of a specific conserved gene? How do the upstream and downstream CRISPR loci look like, in a schematic way, for each system/variant mentioned by the authors (eg. number of DRs, spacer length, etc)? Please modify Figure 1 to provide a better overview of the CRISPR/Cas systems described in this study.
2. In Figure 3, what are the dark-coloured bars? Please clarify. Also, it would be more appropriate to say 3’ end sequences than right end sequences.
3. For the PCA, the authors should perform PERMANOVA to test if there is significant difference between groups. A follow-up heteroscededasticity test using PERMDISP would be necessary to examine whether the ANOVA F statistic could be driven by dispersion differences among groups.
4. Lines 264-335 should be placed under the results section. These are not discussion material.
5. L23: Should be 3,474 genomes. Please use comma instead a period sign. Similar changes are also needed at L111, 112, 182 and 184.
6. L97 and onwards: Please use the dash sign, but not the underscore symbol, when stating the CRISPR-Cas systems.
7. L121 and onwards: We always say upstream or downstream flanking sequence, but not left or right, as the latter does not clearly indicate the direction of the target gene. Please amend.
Author Response
We would like to thank all the reviewers for their suggestions on our manuscript. The answers to the reviewer's questions are detailed further. All the corrections are marked with track changes in the manuscript.
Answers to reviewer comments
Reviewer 2
In the study entitled “ Correlation between CRISPR loci diversity in three enterobacterial taxa”, the authors aimed to examine the diversity of the CRISPR loci present in Klebsiella pneumoniae, Salmonella enterica and Escherichia coli. While the authors stated that the number of CRISPR spacers differed significantly between strains of different geographical origins for each species included in this study, no analysis was further performed to look at the distribution and composition of these CRISPR arrays in different serotypes or phylogroups of these individual species. This would help to better understand the evolution of CRISPR-Cas systems and to explore if CRISPR array could act as a potential molecular marker for any particular ST or phylogroup, improving study significance and the usefulness of the designed primers. Therefore, it is my honest opinion that this study would still require significant amount of work before it can be considered for publication. Please see additional comments below.
The spacers from CRISPR arrays are one of the most hypervariable regions in bacterial genomes. For this reason, they are used in a method called “CRISPR typing” to distinguish between very close-related bacterial isolates, such as in the case of bacterial outbreaks. Most of the previous studies were looking at the composition of the CRISPR array, mostly spacers. In our study on the genomes from databases, we consider that even if they are the same serotype, they could be too phylogenetically distant for the spacers to be used as a molecular marker. We propose a global perspective, and we decided to look at the more conserved part of the arrays, which are the sequences of direct repeats (DRs), that are usually not analysed.
- In the first part of the results section, it is rather difficult to understand the structure and the organisation of the CRISPR-Cas systems present in these three bacterial species and Figure 1 is just too simple. What does it mean that the position relative to the Cas-cluster is the same for all variants? Were these CRISPR/Cas systems always found downstream of a specific conserved gene? How do the upstream and downstream CRISPR loci look like, in a schematic way, for each system/variant mentioned by the authors (eg. number of DRs, spacer length, etc)? Please modify Figure 1 to provide a better overview of the CRISPR/Cas systems described in this study.
In Figure 1, we tried to illustrate as schematic as possible the orientation and the position of the CRISPR loci and Cas cluster relative to each other to show the typical conformation of the systems. For better understanding, we introduced in the description of Figure 1 the definition of CRISPR variants. The spacer length is mentioned in the main text, consisting of 32 N for all loci, and the number of DRs for each locus is represented in Figure 4.
- In Figure 3, what are the dark-coloured bars? Please clarify. Also, it would be more appropriate to say 3’ end sequences than right end sequences.
The description of Figure 3 was rectified to explain what the dark-colored bars represent. We also corrected the expression by replacing the term “right end” with the “3’ end”.
- For the PCA, the authors should perform PERMANOVA to test if there is significant difference between groups. A follow-up heteroscededasticity test using PERMDISPwould be necessary to examine whether the ANOVA F statistic could be driven by dispersion differences among groups.
We agree that the suggested statistical interpretation may better explain the PCA clustering and therefore additional tests were considered.
Results: Further, for all the three species investigated during this study, the multivariate nonparametric permutational analysis of variance (PERMANOVA) revealed significant differences (p < 0.001) between populations by geographical origin, explained by differences in dispersion of CRISPR elements (Welch f-test, p = 0). When assessing the variance between populations by the isolation source, no differences were observed in the variances of CRISPR elements for E. coli and S. enterica. Nevertheless, although the PCA indicates similarity in their CRISPR systems diversity, environmental and human-derived K. pneumoniae strains differ significantly (PERMANOVA, p < 0.001) due to the heterogeneity of variance.
Materials and methods: To assess whether the CRISPR-bearing bacterial populations differ between continents or sources, PERMANOVA was run with 99999 permutations using the PAST software. The assumption of homogenous multivariate dispersion was assessed using the Levene’s test. Under the assumption of heteroscedasticity (Levene’s test), distribution-based Welch f-test was employed to examine if variability is due to dispersion differences among groups.
- Lines 264-335 should be placed under the results section. These are not discussion material.
We agree that lines 264-335 are belonging to results and we placed them accordingly.
- L23: Should be 3,474 genomes. Please use comma instead a period sign. Similar changes are also needed at L111, 112, 182 and 184.
Thank you for the observation, the changes have been made.
- L97 and onwards: Please use the dash sign, but not the underscore symbol, when stating the CRISPR-Cas systems.
The underscore symbol was replaced with the dash sign, as suggested.
- L121 and onwards: We always say upstream or downstream flanking sequence, but not left or right, as the latter does not clearly indicate the direction of the target gene. Please amend.
Thank you, we amended accordingly.
Round 2
Reviewer 2 Report
Thank you for making the changes as per my earlier suggestions. While it might be true that the CRISPR array could not act as a molecular marker to distinguish between serotypes, etc, of the taxa examined in this study, this remains an assumption until proper analyses have been performed. As I do understand that it is difficult to incorporate such analyses into this work as that would require considerable efforts and computational times, and will therefore not insist the need of doing so, but at the very least, I hope the authors would expand their discussion to offer further insights or plausible explanations related to the significant differences observed between regions. For example, the number of CRISPR spacers in the S. enterica genomes of Europe origin was substantially less than those from other continents. What could be the contributing factor to this low number? Is there a dominant subspecies/serovar driving this phenomenon and if there is one, do similar isolates of other geographical origins share comparable spacer profile? This would somewhat give a hint if the CRISPR array could have a potential role as molecular marker. Having an in-depth discussion addressing these observed differences will help boost the value of the findings reported in this study. On a minor note, please replace the "_" sign with a dash like previous in the conclusion as well as Figure 1.
Author Response
Thank you for helping us to improve the quality of the manuscript. We tried to expand our discussions in the suggested direction. The new paragraph was highlighted yellow. The underline sign was also replaced with the dash sign.

Round 3
Reviewer 2 Report
Thank you for the updates. I believe the authors have done their best improving this manuscript. There is no longer any major comment from my side, but am suggesting a few minor changes before the manuscript can be readily accepted for publication. Please check below.
L354-357: “Although the spacer composition had been reported to be correlated with S. enterica serotypes, we are unable to determine such relationship in our work as for most genomes included in this study the serotype information is not provided in the database we used.”
L357-364: delete “Moreover … a high number.”. I would be doubtful if there is no significant difference in the number of spacers between the European and other regional S. enterica isolates. Further, it is not that meaningful or appropriate to compare the spacer distributions among the three different species within any single region. So, I would just suggest removing these statements.
L366-377: For the sentences “However, using specific primers … high efficiency of amplification.”, please consider the following changes.
“Previous studies also developed primers to amplify the CRISPR loci from S. enterica [18,44], E. coli [37,38] and K. pneumoniae [45,46] by targeting the adjacent genes of the system and the cas genes. Since then, a great number of genomes had been added to the public genome repositories for these species. It is important to note that the primers designed in this study were based on a larger number of genome dataset as compared to those from past studies, leading to greater sensitivity and specificity in the amplification of the CRISPR systems in S. enterica, E. coli and K. pneumoniae genomes, in addition to enabling the identification of polymorphism occurring in each system. It would be of our future interest to infer the serotypes and/or sequence types of the bacterial strains using relevant genome-based typing methods prior to extracting the CRISPR systems using our highly efficient primers, to allow further exploration of their potential role as a molecular epidemiological marker.”
L450-456: After another thought, it would be more appropriate to place this paragraph right after L286 in the results section.
Author Response
Thank you for all your suggestions. We rewrote the paragraphs accordingly and we moved the paragraph between L450-456 to the results section, as suggested.
However, since we did not test the efficacy of primers designed in previous studies, we did not want to say that our set of primers is necessarily better. Therefore, we only stated that we worked on a greater number of sequences and that our primers tested in silico were highly efficient, and this aspect we decided not to change.
